

# Simple, reference-independent assessment to empirically guide correction and polishing of hybrid microbial community metagenomic assembly

Garrett J. Smith[1,2], Theo A. van Alen[2], Maartje A.H.J. van Kessel[2] and Sebastian Lücker[2]

[1] Department of Microbiology, The Ohio State University, Columbus, OH, United States of America
[2] Department of Microbiology, Radboud Institute for Biological and Environmental Sciences, Radboud University, Nijmegen, Netherlands

Corresponding authors
Garrett J. Smith,
smith.10284@osu.edu
Sebastian Lücker,
S.Luecker@science.ru.nl

## ABSTRACT

Hybrid metagenomic assembly of microbial communities, leveraging both long- and short-read sequencing technologies, is becoming an increasingly accessible approach, yet its widespread application faces several challenges. High-quality references may not be available for assembly accuracy comparisons common for benchmarking, and certain aspects of hybrid assembly may benefit from dataset-dependent, empiric guidance rather than the application of a uniform approach. In this study, several simple, reference-free characteristics–particularly coding gene content and read recruitment profiles–were hypothesized to be reliable indicators of assembly quality improvement during iterative error-fixing processes. These characteristics were compared to reference-dependent genome- and gene-centric analyses common for microbial community metagenomic studies. Two laboratory-scale bioreactors were sequenced with short- and long-read platforms, and assembled with commonly used software packages. Following long read assembly, long read correction and short read polishing were iterated up to ten times to resolve errors. These iterative processes were shown to have a substantial effect on gene- and genome-centric community compositions. Simple, reference-free assembly characteristics, specifically changes in gene fragmentation and short read recruitment, were robustly correlated with advanced analyses common in published comparative studies, and therefore are suitable proxies for hybrid metagenome assembly quality to simplify the identification of the optimal number of correction and polishing iterations. As hybrid metagenomic sequencing approaches will likely remain relevant due to the low added cost of short-read sequencing for differential coverage binning or the ability to access lower abundance community members, it is imperative that users are equipped to estimate assembly quality prior to downstream analyses.

## BACKGROUND/INTRODUCTION

Though the increasing variety of high-throughput short- and long-read (meta)genomic sequencing technologies are only within their first decades of existence, both the sequencing technologies and software development have flourished and have already been implemented to study microbial ecosystems (*Overholt et al., 2020*; *Singleton et al., 2021*; *Bertrand et al., 2019*; *Liu et al., 2022*; *Ye et al., 2022*; *Ravi et al., 2022*; *Gounot et al., 2022*; *Van Goethem et al., 2021*; *Jin et al., 2022*; *Zhang et al., 2023b*; *Meslier et al., 2022*; *Sereika et al., 2022*; *Stewart et al., 2019*; *Brown et al., 2021*; *Tao et al., 2022*). Integrating both short- and long-read platforms for single microorganisms or microbial communities is gaining popularity because they compensate for the other's weaknesses—shorter reads achieve higher accuracy while longer reads offer better contiguity (*Gounot et al., 2022*; *Meslier et al., 2022*; *Chen et al., 2020*; *Wick & Holt, 2022*; *Zhang et al., 2023a*; *Wick et al., 2017*; *Weirather et al., 2017*). Despite the decreasing costs and improving technologies, hybrid strategies currently are more cost-effective for recovering high-quality metagenome-assembled genomes (MAGs) from microbial communities than either platform alone (*Sereika et al., 2022*). However, remaining biological and technical challenges may deter users from seeking relatively complicated approaches like hybrid (meta)genomic sequencing of microorganisms in complex ecosystems.

Hybrid (meta)genomic assembly implements multiple and/or iterative processes to help overcome the limitations of the sequencing technologies. One strategy is to assemble short reads and then bridge gaps with long reads, leading to assemblies that are accurate but less contiguous (*Overholt et al., 2020*; *Ye et al., 2022*; *Van Goethem et al., 2021*; *Meslier et al., 2022*; *Brown et al., 2021*; *Zhang et al., 2023a*; *Wick et al., 2017*). The second strategy is to assemble long reads and then iteratively use long and short reads to resolve sequencing errors, leading to a more contiguous assembly that may yet retain errors (*Overholt et al., 2020*; *Ye et al., 2022*; *Van Goethem et al., 2021*; *Meslier et al., 2022*; *Brown et al., 2021*; *Wick & Holt, 2022*; *Zhang et al., 2023a*; *Wick et al., 2017*; *Weirather et al., 2017*; *Zimin & Salzberg, 2020*; *Wick et al., 2021*; *Wick & Holt, 2021*; *Wick, Judd & Holt, 2019*; *Watson & Warr, 2019*; *Huang et al., 2021*). In the recent past and near future, as long read accuracy increases tremendously, this second strategy may become overall more favorable for the field because, intuitively, sequence contiguity better enables downstream gene- and genome-resolved analyses. Most benchmarks for the second strategy have provided a general set of guidelines for implementing the iterative processes to improve upon long read assemblies: using several tools is preferred, long read correction prior to short read polishing is advantageous, and iterative processes have diminishing returns (*Liu et al., 2022*; *Sereika et al., 2022*; *Stewart et al., 2019*; *Wick & Holt, 2022*; *Wick et al., 2021*; *Huang et al., 2021*; *Vaser et al., 2017*; *Chen, Erickson & Meng, 2021*; *Damme et al., 2021*; *Belser et al., 2018*; *Hu et al., 2020*; *Mak et al., 2023*; *Zablocki et al., 2021*). These concepts are incorporated into pipelines for automating the optimization of hybrid microbial (meta)genomic assemblies (*Wick et al., 2017*; *Damme et al., 2021*), suggesting they are core aspects of hybrid metagenomic assembly for microorganisms.

Intriguingly, despite its existence in literature essentially since the development of long-read sequencing, the iterative processes for fixing errors in long-read assemblies have been less thoroughly investigated. In particular, one challenge in performing hybrid assembly for a microbial community metagenome is identifying the number of iterations to maximize the quality of the data. Generally, there is little consistency of tool choice and the number of iterations among published studies of microbial communities or other datasets reconstructed using long-read assembly (*Overholt et al., 2020*; *Singleton et al., 2021*; *Bertrand et al., 2019*; *Ye et al., 2022*; *Ravi et al., 2022*; *Gounot et al., 2022*; *Van Goethem et al., 2021*; *Jin et al., 2022*; *Zhang et al., 2023b*; *Meslier et al., 2022*; *Stewart et al., 2019*; *Brown et al., 2021*; *Huang et al., 2021*; *Belser et al., 2018*), and similar inconsistency among benchmarking studies (*Bertrand et al., 2019*; *Meslier et al., 2022*; *Brown et al., 2021*; *Wick & Holt, 2022*; *Zhang et al., 2023a*; *Weirather et al., 2017*; *Zimin & Salzberg, 2020*; *Wick et al., 2021*; *Wick & Holt, 2021*; *Wick, Judd & Holt, 2019*; *Huang et al., 2021*; *Vaser et al., 2017*; *Hu et al., 2020*; *Mak et al., 2023*; *Zablocki et al., 2021*; *Chen, Erickson & Meng, 2020*; *Kolmogorov et al., 2020*; *Koren et al., 2017*; *Firtina et al., 2020*; *Zeng et al., 2020*; *Lee et al., 2021*; *Li, 2016*; *De Maio et al., 2019*; *Antipov et al., 2016*; *Hu et al., 2021*; *Huang et al., 2022*; *Warren et al., 2019*). While most studies have determined the approach *a priori*, some evidence suggests that an unsupervised but empirically-guided approach combining various tools optimizes hybrid microbial assemblies because certain tools are better able to fix certain errors, while others may even degrade the quality by re-introducing errors (*Liu et al., 2022*; *Wick & Holt, 2022*; *Zimin & Salzberg, 2020*; *Huang et al., 2021*; *Hu et al., 2020*). It is likely that ideal hybrid assemblies for microbial communities may not be achieved using a universal or standard protocol but rather may vary depending both on the biology of the system and tool implementation.

An additional challenge of applying hybrid assembly to a complex microbial community is determining the quality of the assembly itself. Typically during benchmarking of long read datasets, assembly qualities are assessed by comparing to high-quality references, *e.g.*, count of differences (mis-assemblies), genomic alignments, or the presence of specific marker genes (*Ye et al., 2022*; *Sereika et al., 2022*; *Zhang et al., 2023a*; *Wick et al., 2021*; *Wick & Holt, 2021*; *Belser et al., 2018*; *Chen, Erickson & Meng, 2020*; *Kolmogorov et al., 2020*; *Lee et al., 2021*; *Dida & Yi, 2021*), which can lead to poor interpretation of consortia or divergent genomes. In the absence of high-quality reference genomes, the characteristics of an ideal hybrid assembly of a complex microbial community are less clear. Common, reference-independent statistics for comparing assemblies, for example contig counts, total base pairs assembled, and L/N50 and similar metrics, might not significantly change at a metagenome assembly scale during iterative processes that fix relatively small-scale errors (*Wick & Holt, 2022*; *Huang et al., 2021*; *Chen, Erickson & Meng, 2021*; *Belser et al., 2018*; *Hu et al., 2020*). Therefore, typical assembly quality assessments are not suitable for many complex microbial community metagenomic datasets.

Benchmarking of individual bacterial genome assemblies have suggested that read recruitment, and to a lesser degree gene counts and/or lengths, are useful reference-free proxy indicators of assembly quality (*Sereika et al., 2022*; *Wick & Holt, 2022*). Supporting their utility, gene calling and read recruitment are simple to generate, analyze, and interpret,

especially as they are integral components of nearly all downstream analyses in gene- and genome-centric studies. In several cases, gene fragmentation and read recruitment profiles, often in comparison to references, have been used to assess recovered MAG quality or the accuracy of assemblies themselves (*Liu et al., 2022*; *Van Goethem et al., 2021*; *Meslier et al., 2022*; *Sereika et al., 2022*; *Stewart et al., 2019*; *Tao et al., 2022*; *Wick & Holt, 2022*; *Zhang et al., 2023a*; *Wick et al., 2021*; *Zablocki et al., 2021*; *Dohm et al., 2020*; *Clark et al., 2013*), but in many cases, there was no apparent evaluation or optimization of the hybrid assembly themselves (*Overholt et al., 2020*; *Singleton et al., 2021*; *Ye et al., 2022*; *Jin et al., 2022*; *Zhang et al., 2023b*; *Meslier et al., 2022*; *Brown et al., 2021*; *Dohm et al., 2020*). Given the challenges of applying hybrid assembly approaches for microbial community metagenomes, and the likelihood that no universal approach works best for all datasets, characteristics that are reference-independent and relative to the dataset may be the most suitable to estimate quality and empirically guide optimization.

Here, we examined multiple reference-independent and -dependent assembly characteristics in order to determine which would be effective for estimating the quality of hybrid metagenomic assemblies of uncharacterized, complex microbial communities, with a focus on the ability to detect changes during iterative error-fixing processes. Two long-term laboratory-scale nitrifying bioreactors were sequenced using both Illumina MiSeq (short-read, SR) and Oxford Nanopore Technology (long-read, LR) platforms and assembled with multiple programs to allow for initial biological and computational variation. LR assemblies were then corrected with LRs and polished with SRs using prevalent tools in the field with low computational demands. Rather than assessing the efficacy of tools, we sought to (1) observe the impact of iterative processes on assembly quality and community reconstruction, and (2) analyze reference-independent assembly characteristics to estimate assembly quality and determine endpoints for correction and polishing. We first established that the fixing of errors during these iterative processes leads to substantial variation in recovered community composition. We then demonstrated that simple, reference-independent assembly characteristics, in particular, coding gene content and/or SR recruitment statistics. Not only did these reference-independent characteristics follow the same patterns within this study and published observations, but also were robustly correlated with reference-dependent characteristics, thus serving as practical proxies for assembly quality and community reconstruction.

## METHODS

The tools, their purpose, and the rationale for or advantages of their use in this study are described in Text S1 and summarized in Table S1.

### Long- and short-read sequencing

Biomass was collected in November 2020 and March 2021 from long-term, autotrophic nitrifying enrichment cultures maintained in either oxygen- (OLR) or nitrogen-limited (NLR) conditions in a tandem laboratory-scale bioreactor system inoculated in 2015 with activated sludge from the Bavaria Brewery wastewater treatment plant in Lieshout, The Netherlands (51.518666 N, 5.613009 E). The cultivation of these bioreactors and the

reconstruction of their microbial communities is an ongoing project and will be published separately. Genomic DNA was extracted from the biomass using a conventional N-cetyl-N,N,N,-trimethyl ammonium bromide (CTAB) protocol in 2020, and the Powersoil DNA Isolation kit (Qiagen, Hilden, Germany) in 2021 with minimal modifications to the manufacturer's directions to reduce DNA shearing such as inverting rather than vortexing or pipetting to mix. Two DNA isolation protocols were used because it is standard in our group due to strong evidence that it introduces sufficient bias to aid differential coverage binning (*Albertsen et al., 2015*; *Weber, De Force & Apprill, 2017*; *Martin-Laurent et al., 2001*).

In total, 1 ng of DNA for both the OLR and NLR reactors was used to prepare a library using the Nextera XT kit (Illumina, San Diego, CA, USA) according to manufacturer's instructions. After quality and quantity check of the libraries, they were paired-end sequenced (2 × 300 bp) using the Illumina MiSeq sequencing machine and the MiSeq Reagent Kit v3 (San Diego, CA, USA) according to manufacturer's protocols. Oxford Nanopore Technologies (ONT) sequencing was done with 840 and 1,670 ng DNA for the OLR and NLR reactors, respectively, after library preparation using the Ligation Sequencing Kit 1D (SQK-LSK108) and the Native Barcoding Expansion Kit (EXP-NBD104), according to the manufacturer's protocols (Oxford Nanopore Technologies, Oxford, UK). The libraries were loaded on a Flow Cell (R9.4.1) and sequenced on a MinION device (Oxford Nanopore Technologies, Oxford, UK), according to the manufacturer's instructions. Guppy (version 4.0.11) (*Oxford Nanopore Technologies, 2023b*) was used to basecall fast5 files using the dna_r9.4.1_450bps_hac.cfg model, both provided by Oxford Nanopore Technologies.

Raw or basecalled sequencing reads for both bioreactors and technologies are available at NCBI *via* BioProject PRJNA1005948 as the following BioSamples: SAMN37004618, raw MiSeq reads for the OLR; SAMN37004620, basecalled ONT reads for the OLR; SAMN37004619, raw MiSeq reads for the NLR; SAMN37004621, basecalled ONT reads for the NLR.

## Long-read, short-read, and hybrid microbial community metagenomic assembly

An overview of the experimental design is shown in Fig. 1. All computational programs were employed with default settings unless explicitly stated. Generic example code is available in File S1.

Sequencing yielded 3.1−4.6 Gbp per library (Table S2) following read trimming and length and quality control using BBduk (BBtools version 37.76) (*Bushnell, 2023*) with a minimum phred score 18 and length of 200 bps for the MiSeq reads, and porechop (version 0.2.3_seqan2.1.1) (*Wick, 2023*) with minimum split length of 3000 bps for the ONT reads.

Three common SPAdes-dependent programs were used to generate SR-only, LR-only, and/or SR-first hybrid (*i.e.,* short read assembly followed by connecting contigs using long read alignments), assemblies: hybridSPAdes (version 3.15.4) (*Antipov et al., 2016*; *Nurk et al., 2017*) known for high-quality metagenomic assemblies, Unicycler (version 0.4.9b) (*Wick et al., 2017*) with utilities for optimizing SPAdes to enable recovery of circular
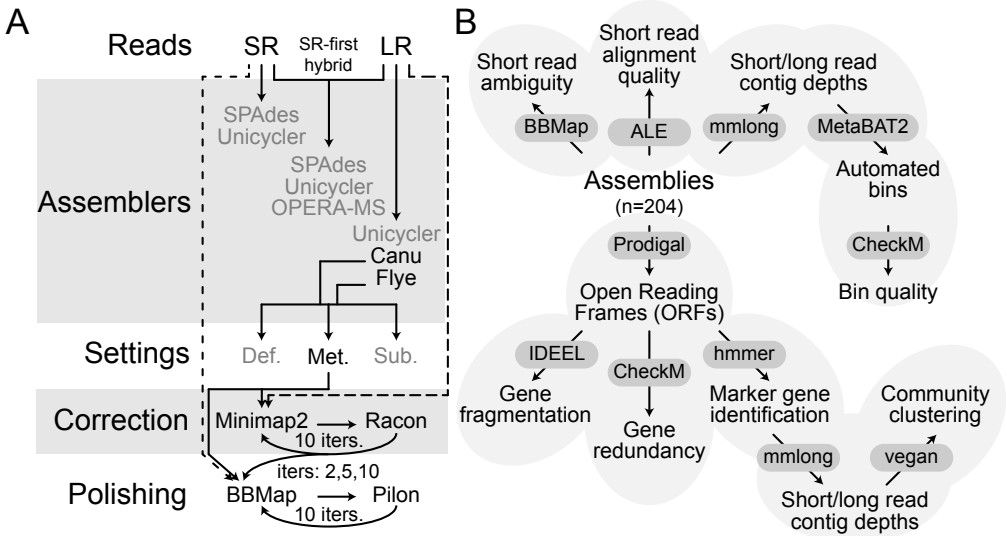

**Figure 1** **Schematic overview of assembly, correction, and polishing.** (A) Assembly, correction, and polishing experimental design. Broken lines represent read recruitment steps for either correction or polishing. Acronyms: SR, short read; LR, long read; Def., default settings; Met., metagenome-optimized settings; Sub., sub-sampled LRs with metagenome-optimized settings. (B) Simplified diagram of analyses performed and the tools used to perform them for each assembly and after each iteration of correction or polishing.

and high-quality single genomes, and OPERA-MS (version 0.9.0) (*Bertrand et al., 2019*) capable of automated refinement of high-quality individual genomes within a multi-species metagenome (hybrid only). Note that Unicycler, when given only long reads, does not use SPAdes and instead uses miniasm (*Li, 2016*) and Racon (*Vaser et al., 2017*). The k-mer list generated by automated selection during Unicycler hybrid assembly was used for hybridSPAdes assemblies. While the focus here was optimizing a hybrid approach that leverages the contiguity of the long-read assemblies, these SPAdes-dependent assemblies served as a baseline for comparing SR and LR assembly of the datasets.

Two programs were used for assembly of LR data: Canu (version 1.8) (*Koren et al., 2017*), which performs read correction prior to assembly and is generally thought to result in more accurate assemblies, and Flye (version 2.9-b1768) (*Kolmogorov et al., 2020*), which performs correction of the assembly using the input reads. Both assemblers were used in three different ways: (1) default settings ("def") developed for single-genomes, (2) metagenomic settings ("meta") to improve assembly of sequences with uneven depths (*Singleton et al., 2021*; *Kolmogorov et al., 2020*), and (3) metagenomic settings for approximately even bp sub-samples ("sub") of the reads to artificially reduce the sequencing depth and possibly uncover biological variation. Long reads were sub-sampled into 12 read pools with approximately the same quantity of bps using the "subsample" utility of Trycycler (version 0.4.1) (*Wick et al., 2021*). All assemblies were filtered to a 4 kbp minimum contig length using BBtools utilities prior to any downstream analyses.

## Long read correction and short read polishing

Rather than identifying the best tool(s) or exact approach, our goal was to demonstrate that each iteration of correction or polishing assembly quality and ability to reconstruct communities. Therefore, we chose only one tool each for LR correction and SR polishing. Consensus correction of individual LR "meta" assemblies was performed using Racon (version 1.3.1) (*Vaser et al., 2017*) for up to 10 iterations, the greatest number of iterations in a hybrid assembly approach that we identified in the literature (*Zablocki et al., 2021*; *Lee et al., 2021*). For each iteration of Racon correction, minimap2 (version 2.17-r941) (*Li, 2018*) was used with default settings to recruit LRs to the original assembly or the previous iteration's corrected assembly to generate the overlap information. Contigs that were not corrected were retained with the optional flag "-u". SR pile-up polishing of LR "meta" assemblies was performed using Pilon (version 1.23) (*Walker et al., 2014*) to 10 iterations after 0 (original assembly), 2, 5, and 10 rounds of LR correction. These stages were chosen to include common (0 and 2) LR correction endpoints found in literature and online guidance, and extensive (5 *Wick et al., 2021* and 10 *Zablocki et al., 2021*; *Lee et al., 2021*; *Zhang et al., 2020*) LR correction endpoints that would help demonstrate the maximum sensible value of this process. Note that many benchmarks, and particularly studies exploring the quality during iterative processes, focus on individual microbial genomes and it thus seemed plausible that additional iterations may aid in improving assembly quality of metagenomic datasets. We did not test SR polishing after 1 iteration of LR correction because it was rarely observed in published literature or online guidelines, and essentially never when Racon was used for LR correction. For each iteration of SR polishing, BBmap (BBtools version 37.76) was used to recruit SRs to the original assembly, Racon-corrected assemblies, or the previous iteration's polished assembly, using a 97.5% identity filter and retaining all ambiguous alignments. As LR correction after SR polishing is expected to re-introduce errors, and we also found no examples in the literature, this was not performed.

## Assembly characteristics determination

Programs for estimating quality were used with default settings except when explicitly stated otherwise. Assemblies were compared using metaQuast (version 5.0.2) (*Mikheenko, Saveliev & Gurevich, 2016*) without references to examine the distribution of contig counts and lengths. Recruitment of SRs to the assemblies to quantify aligned reads and ambiguity was performed using BBmap (BBtools version 37.76) with retaining only perfect mappings of paired reads and randomly assigning ambiguous mappings. The Assembly Likelihood Evaluation (ALE) score of each assembly was determined using the program ALE (*Clark et al., 2013*) (downloaded 2024) after indexing and alignment using BWA (version 0.7.18-r1243-dirty) (*Li & Durbin, 2009*). Open reading frames (ORFs) were predicted using Prodigal (version 2.6.3) (*Hyatt et al., 2010*) using the "meta" procedure for quantification of gene counts and lengths, and as input for downstream analyses of assembled genes. The "meta" procedure of Prodigal was implemented prior to downstream analytical pipelines because it led to increased marker gene recovery estimates, though following similar trends (data not shown), compared to the outputs of the analytical pipelines that assume inputs are
single genomes and therefore run prodigal in "single" mode by default. Phylogenetic marker gene recovery, fragmentation, and redundancy were estimated by Benchmarking Universal Single-Copy Orthologs (BUSCO, version 5.1.2) (*Manni et al., 2021*) using "genome" mode and only the "bacteria_odb10" lineage, as well as CheckM (version 1.1.3) (*Parks et al., 2015*) using the taxonomy workflow for the domain "Bacteria". Gene fragmentation of entire assemblies was estimated by comparing to a DIAMOND (version 0.9.31) (*Buchfink, Xie & Huson, 2015*) database of the uniref50 (downloaded 2023-03-01) (*Suzek et al., 2015*) dataset using IDEEL (downloaded 2023-05-13) (*Stewart et al., 2019*). To more concretely associate potential complete genome contigs with the domain bacteria, circular and long (>1 Mbp) contigs were also analyzed individually using the Genome Taxonomy DataBase toolkit (GTDB-tk, version 1.6.0 with reference database version r202) (*Chaumeil et al., 2020*). Based on the GTDB-tk output for circular contigs alone, there were members of at least six and five distinct bacterial phyla present in the OLR and NLR, respectively (data not shown). Microbiomes with abundant archaea or eukaryotes may need to adjust these taxonomically constrained pipelines to better suit their ecosystem.

## Automated binning and beta diversity

Programs for automated binning and estimation of genome quality, as well as identification and read depth calculation of marker genes, were used with default settings except where noted otherwise.

LRs and SRs from both bioreactor datasets were used for read depth calculation of contigs using the mmlong (version 0.1.2) (*SorenKarst, 2022*) utility *readcoverage2*. Automated binning using composition (*i.e.,* tetranucleotide frequency) and coverage (*i.e.,* read depth) was then performed using Metabat2 (version 2.12.1) (*Kang et al., 2019*). The quality of automated bins was estimated using CheckM (version 1.1.3) (*Parks et al., 2015*), with cutoffs of >50% completeness and <10% contamination scores as "medium quality" (MQ), and >90% completion and <5% contamination as "high quality" (HQ). These oversimplified thresholds are no longer *en vogue* but represent computationally simple, rapid, and bulk estimates compared to contemporary, thorough requirements that include other information like rRNA presence and tRNA counts (*Shaffer et al., 2020*).

The RNA polymerase subunit B (*rpoB*) is a protein-coding gene typically found only once in a genome and is universally conserved among Bacteria, Archaea, and Eukarya. This gene is therefore is tractable to serve as a marker gene for individual species to complement the often poorly assembling 16S rRNA gene in gene-centric phylogenetic analyses of metagenomes. *RpoB* genes were identified in assembled contigs using hmmer (version 3.1b2) (*Eddy, 2011*) with the available model and thresholds for the Protein FAMily (pfam) identifier PF04563.15 (downloaded 2020-06-09). To observe beta-diversity, *rpoB*-containing contigs throughout the iterative correction and polishing processes were compiled for each bioreactor and assembler separately. Then the LRs and SRs from both bioreactor datasets were used for read depth estimation for *rpoB*-containing contigs using the mmlong (version 0.1.2) (*SorenKarst, 2022*) utility *readcoverage2*. The Bray-Curtis abundance-rank dissimilarity between each read set was calculated for subsequent two-axis non-metric multi-dimensional scaling (NMDS) performed using vegan (version 2.5-7)
(*Oksanen et al., 2022*) in R (version 4.1.2) (*R Core Team, 2021*), with the values manually mean-centered and scaled for better comparability. These NMDS analyses did not converge after 50 tries due to low stress, but here the species scores were used to view the clustering and trajectories of the *rpoB*-containing contigs' "species scores" for each assembly rather than the four reads' "site scores".

### Further data analysis and visualization

Logs and outputs were mined for data using various bash commands. Data were ultimately imported into R (version 4.1.2) (*R Core Team, 2021*) for analysis and visualization relying primarily on tidyverse (version 1.3.1) (*Wickham et al., 2019*). Most calculations were additionally normalized to contig length in Mbps to make bioreactors and assemblers more comparable. No specific code was developed for this project due to the focus on applying end-user tools.

## RESULTS

### Single read type and automated hybrid assembly baselines

SR sequencing of the OLR and NLR yielded 7.56 million paired-end reads totaling 4.06 Gbp, and 8.73 million paired-end reads totaling 4.60 Gbp of data, after low-quality read removal and base trimming, respectively. LR sequencing of the OLR and NLR yielded 318 thousand reads totaling 3.13 Gbp, and 416 thousand reads totaling 4.01 Gbp of data, after low quality read removal and base trimming, respectively (Table S2). Reads were either assembled alone (SR-alone or LR-alone) or used as inputs for automated SR-first hybrid assembly. There were substantial differences between the assemblies of the bioreactors, but, more importantly, they also varied across assembly programs and certain settings, including total size, contiguity, and circularity (Text S1, Dataset S1, Figs. S1–S3). These largely recapitulated most benchmarking studies showing that LR assemblies lead to better contiguity, but SR assemblies tend to be more accurate (*Overholt et al., 2020*; *Ye et al., 2022*; *Van Goethem et al., 2021*; *Meslier et al., 2022*; *Wick & Holt, 2022*; *Zimin & Salzberg, 2020*).

From here, we explored the metagenome-optimized LR assemblies over the iterative LR correction and SR polishing iterations to ultimately determine the optimal assembly for these datasets with empirical information. We assessed the microbial composition and its variability using standard gene- and genome-centric analyses like beta-diversity and automated bin recovery. We then further analyzed characteristics that should noticeably change, in contrast to for example the size and contiguity of the assembly, and that ideally would also be reference-free: reduction in gene fragmentation because errors in LR and their assemblies lead to frameshifts that fracture genes, and increase in SR recruitment because these higher-accuracy reads should recruit better to more accurate assemblies. Additionally, we followed automated bin recovery and single phylogenetic marker gene beta diversity to see how well these simple, reference-free characteristics may serve as indicators of community reconstruction. In absence of a "gold standard" for uncharacterized microbial communities, comparisons of the LR correction and SR polishing iterations of the LR assemblies were made to SR-alone and SR-first hybrid assemblies to approximate a comparison to high-quality references that are typically derived from SR datasets.

## Marker gene beta diversity

Many studies of microbial communities applying hybrid metagenomic assembly performed gene-centric analyses (*Singleton et al., 2021*; *Ye et al., 2022*; *Ravi et al., 2022*; *Gounot et al., 2022*; *Van Goethem et al., 2021*; *Zhang et al., 2023b*; *Stewart et al., 2019*; *Brown et al., 2021*), often as a means of inferring metabolic capabilities and therefore ecosystem functions, but also to help access less well-assembled community members to better assess extant diversity. While genome recovery has been examined as a proxy for microbial metagenomic assembly accuracy, there is substantially less effort put into examining the remainder of the community members and how this might impact downstream analyses of the community. To test this, we used the RNA Polymerase subunit B (*rpoB*) gene as a phylogenetic marker for all domains of life for beta-diversity clustering analysis by exploring their SR recruitment profiles *via* non-metric multi-dimensional scaling (NMDS).

The error-fixing processes clearly affected the trajectories of community composition by increased *rpoB* recovery and changes to read depth profiles. The recovery and read recruitment of *rpoB* was impacted by the assembler program, as well as the number of LR correction and SR polishing iterations, generally showing an increase in *rpoB* recovery primarily *via* SR polishing (Text S1, Dataset S1, Fig. S4). Even with both LR correction and SR polishing, fewer *rpoB* genes were recovered from LR assemblies than SR-alone or SR-first hybrid assemblies. However, their gene lengths were comparable suggesting that these genes generally assembled and polished well (Text S1, Dataset S1, Fig. S5). There was an unexpected trend of reduced mean depth with LR correction and polishing (Text S1, Dataset S1, Fig. S6), which was likely due to the recovery of genes that were initially fragmented in the assemblies of lower abundance community members. The clustering of assemblies with only LR correction were noisy and remained distant from one another, compared to the convergence observed due to the first few SR polishing iterations (Fig. 2, Dataset S1). In most cases, assemblies without LR correction were distant from assemblies with both LR correction and SR polishing, highlighting the beneficial impact of LR correction that may only become apparent after SR polishing (Fig. 2, Dataset S1). Unfortunately, while communities converged during SR polishing, they remained somewhat distinct depending on the number of LR correction iterations, indicating that LR correction impacts community reconstruction even with SR polishing (Fig. 2, Dataset S1). These results show that iterative LR correction and SR polishing processes together are vital for gene-centric community reconstruction, and thus their downstream analyses.

## Automated bin recovery

Complementing gene-centric approaches, the primary goal for many microbial community metagenomic sequencing approaches is to reconstruct high-quality, ideally essentially complete genomes of as many members as possible (*Overholt et al., 2020*; *Singleton et al., 2021*; *Bertrand et al., 2019*; *Liu et al., 2022*; *Ye et al., 2022*; *Ravi et al., 2022*; *Gounot et al., 2022*; *Van Goethem et al., 2021*; *Jin et al., 2022*; *Zhang et al., 2023b*; *Sereika et al., 2022*; *Stewart et al., 2019*). Hybrid assembly of microbial metagenomes has already been employed to recover microbial genomes from a variety of ecosystems from plant-rich

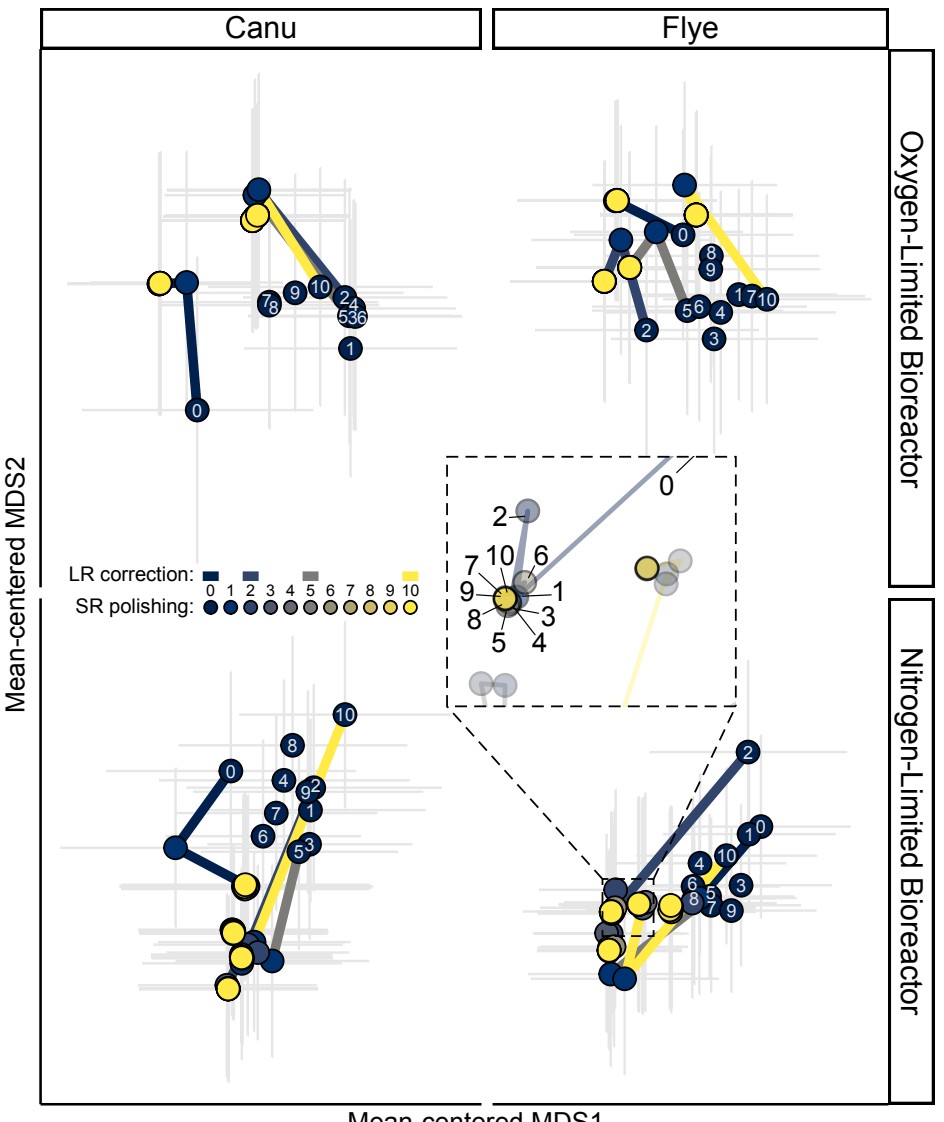

**Figure 2** Beta diversity estimated from RNA Polymerase subunit B (*rpoB*) gene profiles for each bioreactor and long read (LR) assembly throughout the iterative LR correction and short read (SR) polishing processes. The two bioreactors are separated over vertical panels, the two LR assemblers are separated over the horizontal panels. Points are the mean-centered species scores for all *rpoB* genes per assembly, with gray lines showing the standard errors of the mean. Each point is colored by the SR polishing iteration, with colored lines connecting points to the preceding LR correction and subsequent SR polishing iteration. White text over points indicates the LR correction iteration prior to any SR polishing. Inset shows the overlapping points of the second LR correction stage as the communities converge due to SR polishing for the Nitrogen-Limited Bioreactor's Flye assembly.

sediments to waste sludge, to human guts, and more (*Overholt et al., 2020*; *Singleton et al., 2021*; *Bertrand et al., 2019*; *Liu et al., 2022*; *Ye et al., 2022*; *Ravi et al., 2022*; *Gounot et al., 2022*; *Van Goethem et al., 2021*; *Jin et al., 2022*; *Zhang et al., 2023b*; *Sereika et al., 2022*; *Stewart et al., 2019*). To maximize the value of recovered genomes, the assembly itself needs

to be accurate because both the binning of contigs into genomes and the subsequent bin quality estimation *via* marker gene identification depend on the accuracy of the assembly. In some cases, genome recovery has already been used to demonstrate or compare the accuracy of LR and/or hybrid assemblies of microbial (meta)genomes (*Overholt et al., 2020*; *Gounot et al., 2022*; *Zhang et al., 2023b*; *Tao et al., 2022*; *Zhang et al., 2023a*; *Kolmogorov et al., 2020*). Assemblies were automatically binned at each stage of LR correction and SR polishing, and both the assemblies and bins were assessed with CheckM to estimate the total redundancy of the assemblies and the completion of microbial genomes.

Similar to the marker gene beta-diversity analysis, iterative LR correction and SR polishing processes impacted genome recovery, but was noisier. For the initial assemblies and throughout LR correction, the count and quality of medium quality (MQ) or better bins were lower than after SR polishing (Fig. 3, Text S1, Dataset S1, Fig. S7). In contrast to *rpoB* recovery, LR assemblies with LR correction and SR polishing yielded more medium- and high-quality bins than SR-alone or SR-first hybrid assemblies (Dataset S1, Fig. S8). Automated bin recovery metrics remained somewhat noisy and often never reached a clear plateau (Fig. 3, Text S1, Dataset S1, Fig. S7), suggesting that genome recovery may not be a robust assembly quality indicator. The saturation patterns in binning mostly mirrored assembly-level marker gene copy recovery, as well as total redundancy, *i.e.*, sum of completeness and contamination scores, which was also predictive of MQ bin recovery (adjusted $R^2 \geqq 0.85$, $p << 0.05$, Text S1, Fig. S9). While the slope ($\sim 0.01$) indicated an approximately 100:1 relationship between total redundancy and MQ genome recovery, the distance from the idealized line (*i.e.*, $y$-intercept of 0, Fig. S9) indicated that the recovery of genomes representing all community members was unlikely, even from this hybrid assembly strategy. In summary, like marker gene beta-diversity, the integration of LR correction and SR polishing of LR assemblies is crucial for maximizing the yield of quality microbial genomes.

## Gene fragmentation

Small-scale errors (*i.e.*, insertions, deletions, and, to a lesser extent, substitutions) in LRs are known issues that cause gene fragmentation in the resulting assemblies (*Gounot et al., 2022*; *Sereika et al., 2022*; *Stewart et al., 2019*; *Wick & Holt, 2022*; *Weirather et al., 2017*; *Wick & Holt, 2021*; *Huang et al., 2021*; *Belser et al., 2018*; *Hu et al., 2020*; *Dohm et al., 2020*; *Walker et al., 2014*; *Amarasinghe et al., 2020*). Theoretically, as errors are fixed during the iterative LR correction and/or SR polishing processes, then gene fragmentation should decrease, resulting in fewer genes with more bps within them. In benchmarking studies, this is often indirectly measured by comparing differences in gene counts or marker gene inventories (*i.e.*, completion scores), and some have already demonstrated that directly assessing gene lengths works as a reasonable proxy for the accuracy of hybrid assemblies of microbial (meta)genomes (*Liu et al., 2022*; *Gounot et al., 2022*; *Meslier et al., 2022*; *Sereika et al., 2022*; *Stewart et al., 2019*; *Wick & Holt, 2022*; *Zhang et al., 2023a*; *Wick et al., 2021*; *Lee et al., 2021*; *Dohm et al., 2020*). As a precursor to many downstream analyses, coding gene identification with programs like Prodigal allows direct estimates of coding gene lengths, while programs like IDEEL and BUSCO can estimate gene fragmentation by

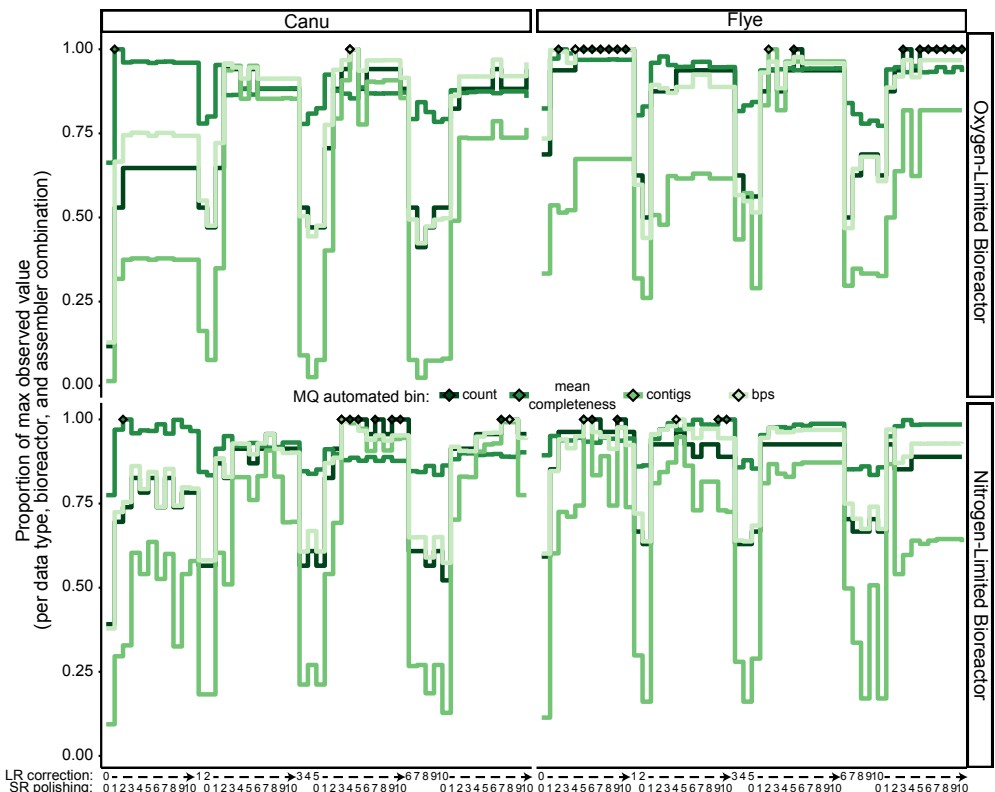

**Figure 3** **Stair plots of automated medium-quality or better (MQ) bin recovery for each bioreactor and long read (LR) assembly throughout the iterative LR correction and short read (SR) polishing processes.** The two bioreactors are separated over vertical panels, the two LR assemblers are separated over the horizontal panels. The LR correction and SR polishing iterations are spread across the $x$-axis so that the ten SR polishing steps are immediately to the right of the respective preceding LR correction step. Each colored line represents the proportion of the maximum observed value per combination of bioreactor and assembler of one of the following characteristics MQ automated bins: the number of MQ bins recovered ("count"), the mean completeness score of the bins estimated by CheckM ("mean completeness"), the number of contigs in MQ bins ("contigs"), and the quantity of bps in MQ bins ("bps"). Matching color diamonds indicate the LR correction and/or SR polishing stage with the maximum observed value for each characteristic.

comparison to reference databases and are commonly used by benchmarking or other comparative studies (*Liu et al., 2022*; *Sereika et al., 2022*; *Stewart et al., 2019*; *Wick et al., 2021*; *Belser et al., 2018*; *Lee et al., 2021*; *Krakau et al., 2022*).

Gene fragmentation substantially decreased and plateaued during iterative error-fixing processes. Analysis of coding genes showed substantial improvements in gene counts and lengths, particularly in response to SR polishing (Text S1, Dataset S1, Fig. S10). This was apparent for the entire assembly, and, to an even greater extent, circular and long contigs that may represent essentially complete microbial sequences, and automated MQ bins (Text S1, Dataset S1, Figs. S10 and S11). These patterns were matched by reductions in fragmented marker genes found by BUSCO (Text S1, Dataset S1, Fig. S11), and were robustly correlated to the proportion of coding genes within 5% of their nearest reference

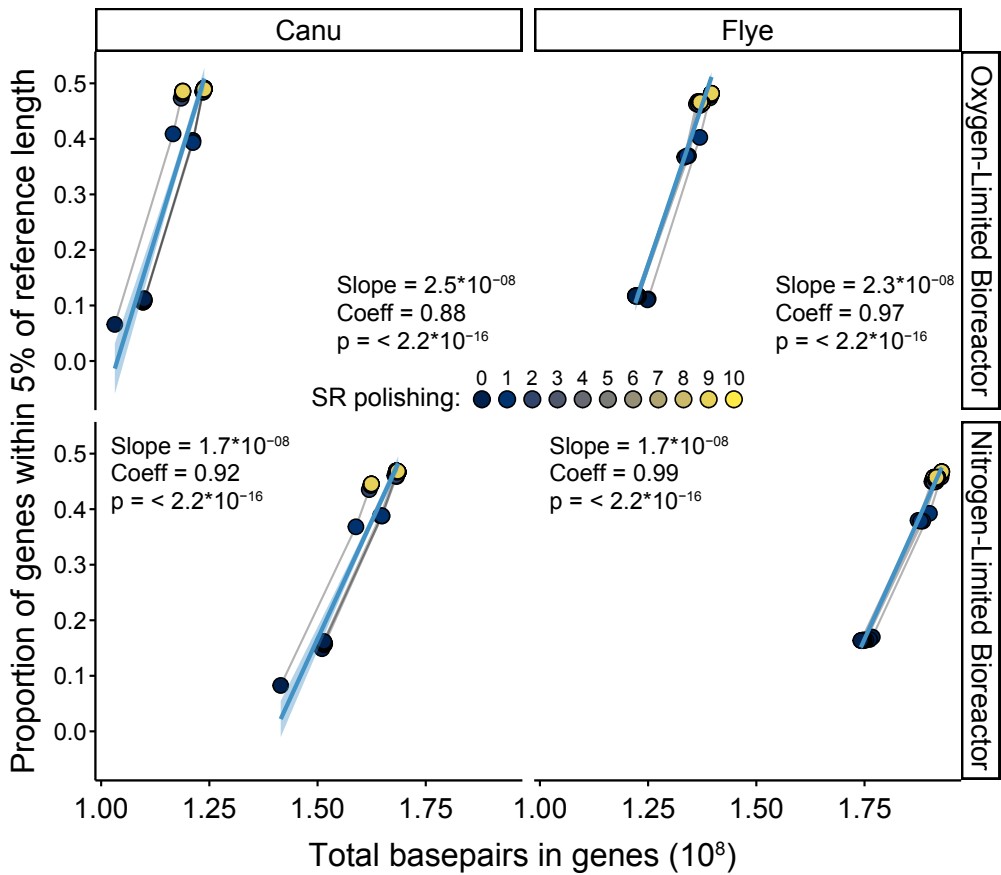

**Figure 4** Linear correlation between calculated bps in genes and the proportion of genes on sufficiently deeply (>1x) sequenced contigs estimated to be similar length to reference database entries using IDEEL for each bioreactor and long read (LR) assembly through. The two bioreactors are separated over vertical panels, and the two LR assemblers are separated over the horizontal panels. Each point is colored by the SR polishing iteration, with lines connecting the points from the same LR correction iteration, all of which are partially transparent. The light blue solid lines and shaded regions are the linear regressions for the displayed data and their 95% confidence intervals. Slopes, correlation coefficients (adjusted $R^2$) and *p*-values are displayed in the upper-left corner of each panel.

length determined using IDEEL (adjusted $R^2 \geqq 0.94$, p-adj $<< 0.05$, Text S1, Dataset S1). However, the fraction of these genes was low and reached only ~25% of the total coding genes (Text S1, Dataset S1, Fig. S12). The removal of contigs with low SR depth (<1x) drastically improved this to ~50% (Text S1, Dataset S1), which was on par with SR-alone and SR-first hybrid assemblies and remained robustly correlated with IDEEL scores (adjusted $R^2 \geqq 0.94$, p-adj $<<0.05$, Fig. 4, Text S1, Dataset S1, Fig. S13). As expected based on both the theory behind the iterative error-fixing processes and most published studies assessing this information (*Liu et al., 2022*; *Meslier et al., 2022*; *Sereika et al., 2022*; *Stewart et al., 2019*; *Wick & Holt, 2022*; *Zhang et al., 2023a*; *Wick et al., 2021*; *Lee et al., 2021*; *Dohm et al., 2020*), gene fragmentation was improved primarily due to the first few iterations of

SR polishing. However, we show here the feasibility of estimating this from coding gene contents alone without the requirement of comparison to reference sequences.

### Short read recruitment

Multiple iterations of LR correction and SR polishing are often performed in benchmarks and published studies with the expectation that newly fixed errors allow different read pools to align. Theoretically, SR recruitment will reach a stable maximum as the assembly approaches high accuracy because fewer errors are fixed in each iteration, leading to fewer changes in read alignment. Several studies have already shown a relationship between microbial (meta)genome quality and read recruitment (*Liu et al., 2022*; *Meslier et al., 2022*; *Sereika et al., 2022*; *Tao et al., 2022*; *Wick & Holt, 2022*; *Wick et al., 2021*; *Dohm et al., 2020*; *Clark et al., 2013*). For example, ALE was developed to use SR alignments to assess assembly quality, though its negative log-likelihood outputs are not intuitive and limits comparability between studies. In contrast, other SR alignment information, for example the proportion of aligned SR, total SR bps recruited, or SR ambiguity, is generally intuitive to interpret and may be comparable between studies.

Consistent with expectations and gene fragmentation results, SR recruitment increased greatly due to SR polishing and then saturated after the first few iterations. While LR correction caused little change, there were substantial improvements during SR polishing including the proportion of total reads and bps, total read count and bps, read ambiguity, and contigs lacking aligned SR (Fig. 5, Dataset S1, Text S1). These results were consistent with many studies that have demonstrated that the greatest improvements occur within the first few iterations of SR polishing, with at least two rounds necessary (*Singleton et al., 2021*; *Jin et al., 2022*; *Zhang et al., 2023b*; *Meslier et al., 2022*; *Sereika et al., 2022*; *Stewart et al., 2019*; *Tao et al., 2022*; *Wick & Holt, 2022*; *Wick et al., 2021*; *Damme et al., 2021*; *Kolmogorov et al., 2020*; *Dohm et al., 2020*; *Walker et al., 2014*). Furthermore, the proportion of total SR bps recruited was robustly correlated with the inverse relative ALE score (adjusted $R^2 \geqq 0.87$, p-adj $<<0.05$, Dataset S1, Fig. S14), indicating that SR recruitment alone was predictive of assembly quality. Similar to gene fragmentation, the ALE scores were drastically improved by the removal of contigs with low SR depth (<1x), which resulted in SR-polished LR assemblies achieving raw ALE scores similar to SR-alone and SR-first hybrid assemblies (Dataset S1, Fig. S15). Overall, SR polishing led to the greatest improvements in SR recruitment, and simple SR alignment statistics appear to be reliable indicators of assembly quality without the requirement of further complex computations.

## DISCUSSION

### Correction and polishing iterations affect gene- and genome-centric community reconstructions

Hybrid assembly approaches, leveraging the complementary, beneficial attributes of both LR and SR sequencing platforms to overcome their limitations, are already being used to study microbial communities in various environments (*Overholt et al., 2020*; *Singleton et al., 2021*; *Bertrand et al., 2019*; *Liu et al., 2022*; *Ye et al., 2022*; *Ravi et al., 2022*; *Gounot et al., 2022*; *Van Goethem et al., 2021*; *Jin et al., 2022*; *Zhang et al., 2023b*; *Meslier et al.,*

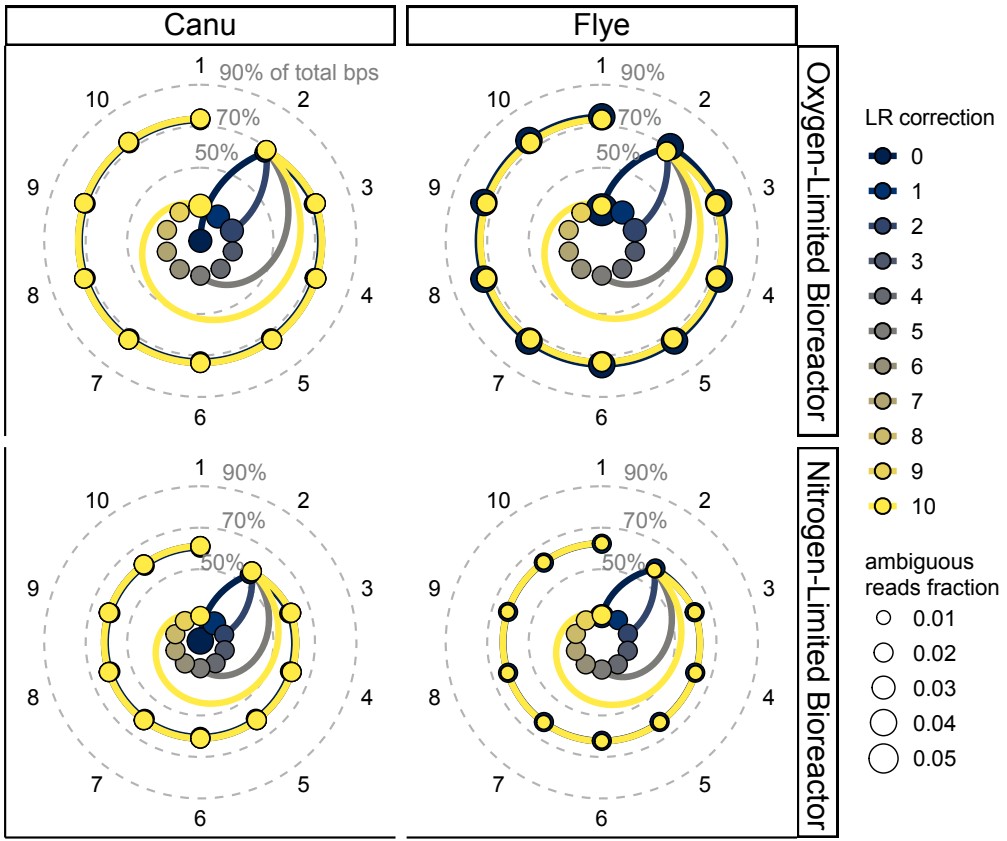

**Figure 5** **Polar coordinate plot of short read (SR) recruitment for each bioreactor and long read (LR) assembly throughout the iterative LR correction and SR polishing processes.** The two bioreactors are separated over vertical panels, and the two LR assemblers are separated over the horizontal panels. Each panel shows a polar coordinate plot of the LR correction or SR polishing iteration in a circle with the percentage of total bps in recruited SR as the distance from the center. The innermost ten points (due to lower values) indicate the percentage of total bps in recruited SR after up to ten LR correction iterations without SR polishing. The colored lines connecting points of the same color link assemblies with the same number of LR correction steps preceding SR polishing. The size of the points indicates the fraction of SR that were ambiguously recruited, *i.e.,* SR aligning to 2 or more position.

*2022*; *Sereika et al., 2022*; *Stewart et al., 2019*; *Brown et al., 2021*; *Tao et al., 2022*). Due to the ability to achieve greater contiguity, assembly of LRs offers a major advantage over SR data alone. However, it comes at the cost of lower accuracy and necessitates the use of additional steps to improve the reliability of the assemblies (*Overholt et al., 2020*; *Ye et al., 2022*; *Van Goethem et al., 2021*; *Meslier et al., 2022*; *Brown et al., 2021*; *Wick & Holt, 2022*; *Zimin & Salzberg, 2020*; *Wick et al., 2021*; *Wick & Holt, 2021*; *Wick, Judd & Holt, 2019*; *Chen, Erickson & Meng, 2020*). Most studies performing benchmarking or comparison of community reconstructions applied different methods in their hybrid assembly approaches, and the biological and technical complexity of microbial community metagenomes probably limits the implementation of uniform methods across different studies. The iterative process of LR correction and SR polishing to improve the quality of

LR assemblies is one of the most inconsistently applied methods during hybrid assembly, and is likely both dataset- and tool-dependent.

Few studies have explored the impacts of these iterative processes, despite being implemented by most of them. Generally, the consensus appears to be that while SR polishing offers the greatest improvements, some LR correction is advantageous, and both offer diminishing returns with increasing iterations (*Stewart et al., 2019*; *Wick & Holt, 2022*; *Zimin & Salzberg, 2020*; *Huang et al., 2021*; *Hu et al., 2020*; *Mak et al., 2023*; *Zablocki et al., 2021*; *Lee et al., 2021*). Substantial amounts of changes still occur even after 10 iterations of SR polishing (Fig. S16), but it is unclear if these changes are beneficial or detrimental, possibly resulting in collapsing of real variation. However, it has not been clearly demonstrated that they affect community reconstruction, and therefore also impact downstream microbial metagenome analysis and interpretation. Here, we first showed that beta diversity clustering substantially changed during these iterative processes, which was, as far as we know, the first demonstration of this phenomenon. We further showed that genome recovery was also impacted by these processes. Together, these demonstrated that endpoints of these LR correction and SR polishing have a significant impact on community reconstruction from metagenomic datasets. Fortunately, the results here followed similar patterns with each other, and also mirrored published studies comparing different aspects of hybrid assembly across multiple tools and over several iterations (*Overholt et al., 2020*; *Singleton et al., 2021*; *Liu et al., 2022*; *Gounot et al., 2022*; *Zhang et al., 2023b*; *Sereika et al., 2022*; *Stewart et al., 2019*; *Brown et al., 2021*; *Tao et al., 2022*; *Wick & Holt, 2022*; *Zhang et al., 2023a*; *Wick et al., 2021*; *Huang et al., 2021*; *Vaser et al., 2017*; *Chen, Erickson & Meng, 2021*; *Damme et al., 2021*; *Belser et al., 2018*; *Hu et al., 2020*; *Mak et al., 2023*; *Zablocki et al., 2021*; *Lee et al., 2021*). However, there is not only substantial methodological variation between approaches published and performed here, but also biological variation between each dataset which limits the application of a standardized methodology for hybrid metagenomic assembly of microbial communities. Therefore, we posit that each dataset needs to be empirically evaluated to determine the optimal hybrid assembly approach.

## Simple, reference-independent assembly characteristics as proxies for the quality of hybrid assemblies of microbial community metagenomes

Hybrid assembly of microbial community metagenomes is complicated by the lack of a clear assessment strategy for assembly quality. The most common assembly quality metrics include estimates of mis-assemblies and contiguity, but these characteristics either require high-quality reference genomes that may not be available for poorly characterized ecosystems, or would not change during LR correction or SR polishing (*Ye et al., 2022*; *Sereika et al., 2022*; *Wick & Holt, 2022*; *Zhang et al., 2023a*; *Wick et al., 2021*; *Wick & Holt, 2021*; *Huang et al., 2021*; *Vaser et al., 2017*; *Chen, Erickson & Meng, 2021*; *Belser et al., 2018*; *Hu et al., 2020*; *Chen, Erickson & Meng, 2020*; *Kolmogorov et al., 2020*; *Lee et al., 2021*; *Dida & Yi, 2021*). We note that in a clustering analysis of the results from commonly used tools, assemblies grouped by the strategy used to generate them as well as the analyses themselves (Dataset S1, Fig. S17). SR-first and LR-first hybrid assemblies migrate towards each other,

*i.e.,* become more similar in these statistical snapshots, which appears largely driven by the results from analysis of automated bin recovery, gene fragmentation, and SR recruitment analyses. Therefore, assembly characteristics appropriate for a microbial community may need to be both dataset- and assembly strategy-dependent.

Here, to help optimize hybrid assembly of uncharacterized microbial community metagenomes, several assembly characteristics were tracked during the iterative processes of LR correction and SR polishing to help determine suitable proxies for assembly quality. Specifically, gene fragmentation and SR recruitment were focused on because these features have already been proposed as suitable proxies for hybrid assembly of microbial genomes (*Liu et al., 2022*; *Meslier et al., 2022*; *Sereika et al., 2022*; *Stewart et al., 2019*; *Wick & Holt, 2022*; *Wick et al., 2021*; *Zablocki et al., 2021*; *Dohm et al., 2020*). These simple characteristics followed patterns consistent with many benchmarking and comparative studies (*Liu et al., 2022*; *Sereika et al., 2022*; *Stewart et al., 2019*; *Wick & Holt, 2022*; *Wick et al., 2021*; *Huang et al., 2021*; *Vaser et al., 2017*; *Chen, Erickson & Meng, 2021*; *Damme et al., 2021*; *Belser et al., 2018*; *Hu et al., 2020*; *Mak et al., 2023*; *Zablocki et al., 2021*; *Zablocki et al., 2021*): (1) LR correction improved the accuracy of the assemblies, which was not always apparent until after SR polishing; (2) the greatest improvements occurred within the first three iterations of SR polishing; (3) beyond five iterations, neither LR correction nor SR polishing offered observable improvements. Furthermore, the number of coding genes and the bps within them were robustly correlated with profiles produced by IDEEL, and SR recruitment was robustly correlated with ALE scores, which demonstrated their value as assembly quality proxies without the need for further computational analyses. Additionally, coding gene and SR recruitment characteristics followed similar patterns as automated bin recovery and marker gene beta diversity trajectories, and therefore are suitable to estimate the point at which community reconstruction will not change substantially between iterations, representing appropriate endpoints of the iterative correction and polishing processes. Thus, these simple, reference-independent assembly characteristics are reliable means to assessing assembly accuracy and can be used to empirically optimize hybrid assemblies of uncharacterized microbial community metagenomes.

## Unexplored factors and limitations of this study

The approach in this study was not without issue, and we therefore describe expected deviations from our approach that may be necessary for certain users or datasets.

First, while recent technological advances in LR sequencing will ease LR integration into hybrid metagenomic assembly of microbial communities, this study was performed using older sequencing chemistry and basecalling algorithms. Some differences remain, but both major LR platforms, Pacific Biosciences (PacBio) and Oxford Nanopore Technologies (ONT) have achieved large improvements in chemistry and basecalling that have increased sequencing depth and accuracy (*Liu et al., 2022*; *Meslier et al., 2022*; *Sereika et al., 2022*; *Wick, Judd & Holt, 2019*; *Dohm et al., 2020*; *PacBioRevio, 2023*; *Oxford Nanopore Technologies, 2023a*). As a result of these LR sequencing advances, LR-alone assemblies can be sufficient for retrieving HQ genomes without SR polishing (*Sereika et al., 2022*; *Zhao et al., 2023*). However, resources may restrict users to older and more

accessible technologies, or projects may already have older datasets that still need analyses integrating short- and long-read technologies. Additionally, for the foreseeable future, SR complements to LR datasets will likely continue to increase the sample count for differential coverage binning. Furthermore, SRs still appear to aid assembly accuracy for high-quality microbial genomes reconstructed from LR (*Sereika et al., 2022*; *Zhang, Jain & Aluru, 2020*). Therefore, combining both sequencing technologies will remain attractive as a cost-effective approach yielding the highest quality data compared to any single platform (*Sereika et al., 2022*).

Second, the myriad of programs and pipelines to process LRs, including basecalling, read correction, assembly, assembly correction, SR recruitment and polishing, and more, continuously improve and expand (*Bertrand et al., 2019*; *Liu et al., 2022*; *Brown et al., 2021*; *Wick & Holt, 2022*; *Zhang et al., 2023a*; *Zimin & Salzberg, 2020*; *Wick et al., 2021*; *Wick & Holt, 2021*; *Wick, Judd & Holt, 2019*; *Huang et al., 2021*; *Vaser et al., 2017*; *Damme et al., 2021*; *Hu et al., 2020*; *Mak et al., 2023*; *Kolmogorov et al., 2020*; *Koren et al., 2017*; *Firtina et al., 2020*; *Zeng et al., 2020*; *Lee et al., 2021*; *Li, 2016*; *Antipov et al., 2016*; *Hu et al., 2021*; *Huang et al., 2022*; *Warren et al., 2019*; *Dohm et al., 2020*; *Oxford Nanopore Technologies, 2023b*; *Nurk et al., 2017*; *Walker et al., 2014*; *Zhang et al., 2020*; *Amarasinghe et al., 2020*; *Krakau et al., 2022*; *Kundu, Casey & Sung, 2019*; *Shafin et al., 2021*; *Oxford Nanopore Technologies, 2023c*; *Huang, Liu & Shih, 2021*; *Ruan & Li, 2020*; *Shafin et al., 2020*; *Vaser & Šikić, 2021*; *Chen et al., 2021*; *Hu et al., 2023*; *Pagès-Gallego & De Ridder, 2023*; *Konishi et al., 2021*; *Xu et al., 2021*; *Lv et al., 2020*; *Miculinić, Ratković & Šikić, 2019*). To reasonably perform this study and analyze the results, only two assembly, one LR correction, and one SR polishing programs were tested, thus introducing technical variation during assembly but limiting it during LR correction and SR polishing. These programs were chosen largely based on their observed prevalence in literature and online resources, but we acknowledge that, in particular, Racon and Pilon may be outcompeted by others that yield better results (*Wick & Holt, 2022*; *Zimin & Salzberg, 2020*; *Huang et al., 2021*; *Hu et al., 2020*; *Mak et al., 2023*; *Firtina et al., 2020*; *Lee et al., 2021*; *Hu et al., 2021*; *Warren et al., 2019*; *Kundu, Casey & Sung, 2019*; *Shafin et al., 2021*; *Oxford Nanopore Technologies, 2023c*; *Huang, Liu & Shih, 2021*; *Ruan & Li, 2020*). Still, the programs we used are also dataset agnostic, and have relatively minimal computational requirements in contrast to tools that are dataset-dependent or demand advanced/specific computational capacities, making them a relatively universal option. For several available LR or hybrid assembly programs, Racon or Pilon are already implemented (*Wick et al., 2017*; *Huang et al., 2021*; *Damme et al., 2021*), indicating their value to the field even though other tools or combinations may yield more accurate assemblies. We further note that each of these computational tools also comes with a suite of settings to fine-tune performance that were not explored here.

Third, we did not separate contig pools for some analyses in order to simplify the workflow. Specifically, in this workflow large circular or linear contigs that likely constituted nearly complete genomes were not removed from the assemblies prior to LR correction or SR polishing. This simplification might result in the introduction of errors, derived particularly from SR that ambiguously align to conserved regions within different strains. However, SR ambiguity decreased during the iterative error-fixing processes, presumably

due to the availability of multiple, more accurate sequences to differentially recruit ambiguous reads after each iteration of SR polishing.

Fourth and lastly, we acknowledge that time points of biomass collection for DNA isolation and sequencing with the different platforms were separated by several months. Though largely stable after enrichment for over five years, the communities in these bioreactors still shift slowly, likely largely due to strains competing for the same niches, which will be examined in follow-up studies. These differences could impact ideal integration of the LR and SR datasets if strains substantially shifted in abundance, leading to improper error-fixing events and possibly producing inaccurate consensus genomes rather than a realistic strain representation.

## CONCLUSIONS

Integrating LR and SR sequencing platforms for hybrid assembly of microbial community metagenomes is challenging due to both biological and technical complexities. Benchmarks and comparative studies seeking to maximize high-quality data yields by combining these technologies have led to some consensus in the approach, but it is possible that direct replication of the methods may not serve all datasets equally well. More specifically, here we have shown that the iterative process of resolving errors in LR assemblies has a substantial impact on the community reconstruction, which may not have been observed, or even observable, in benchmarks. Additionally, we have demonstrated that coding gene contents and SR recruitment are simple, reference-free assembly characteristics that are both sensitive to changes made during iterative correction and polishing process. Furthermore, these can serve as reliable indicators of community reconstruction ability because they were also robustly correlated to complex, reference-dependent analyses of assembly quality. Rather than informing the field on the "best" approach for all datasets, we encourage users correcting and/or polishing LR assemblies to use coding gene contents (counts, lengths, or bps in them) and/or SR recruitment (proportion of total reads, total bps in reads, or ambiguous alignments) to help empirically determine the optimal endpoints of the iterative processes.

### Funding
This work was supported by the Netherlands Organisation for Scientific Research (NWO; Gravitation Grant SIAM 024.002.002, 016.Veni.192.062 and 016.Vidi.189.050). The funders had no role in study design, data collection and analysis, decision to publish, or preparation of the manuscript.

### Grant Disclosures
The following grant information was disclosed by the authors:
Netherlands Organisation for Scientific Research: SIAM 024.002.002, 016.Veni.192.062, 016.Vidi.189.050.

## Competing Interests

The authors declare there are no competing interests.

## Author Contributions

- Garrett J. Smith conceived and designed the experiments, performed the experiments, analyzed the data, prepared figures and/or tables, authored or reviewed drafts of the article, and approved the final draft.
- Theo A. van Alen performed the experiments, authored or reviewed drafts of the article, and approved the final draft.
- Maartje A.H.J. van Kessel conceived and designed the experiments, performed the experiments, authored or reviewed drafts of the article, maintained bioreactors and performed preliminary analyses, and approved the final draft.
- Sebastian Lücker conceived and designed the experiments, authored or reviewed drafts of the article, and approved the final draft.

## DNA Deposition

The following information was supplied regarding the deposition of DNA sequences:

The raw sequence reads are available at NCBI BioProject PRJNA1005948.

## Data Availability

The example code for the parameters and implantation of various end-user bioinformatic tools is in the Supplementary File.

## Supplemental Information

Supplemental information for this article can be found online at http://dx.doi.org/10.7717/peerj.18132#supplemental-information.

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
