# Peer review of "Simple, reference-independent assessment to empirically guide correction and polishing of hybrid microbial community metagenomic assembly"

_PeerJ, doi:10.7717/peerj.18132_

## Round 0.1 · original submission · Major Revisions

Dear Dr. Smith and colleagues:

Thanks for submitting your manuscript to PeerJ. I have now received three independent reviews of your work, and as you will see, the reviewers raised some concerns about the research. Despite this, these reviewers are optimistic about your work and the potential impact it will have on research studying metagenome assembly. Thus, I encourage you to revise your manuscript, accordingly, considering all of the concerns raised by the three reviewers.

The reviewers found problems with your benchmarking, with two reviewers feeling that for benchmarking type of paper the experimental design is flawed; e.g., no ground truth is known, so there might be biases due to heuristic evaluation tools being used. Also, only single polishing tool was used (that is not quite state-of-the-art). Another study was suggested for referencing: https://doi.org/10.1038/s41598-021-00178-w.

There are many specific issues pointed out by the reviewers, and you will need to address all of these and expect a thorough review of your revised manuscript by these same reviewers.

Therefore, I am recommending that you revise your manuscript, accordingly, taking into account all of the issues raised by the reviewers.

I look forward to seeing your revision, and thanks again for submitting your work to PeerJ.

Best,

-joe

·

Basic reporting

This manuscript discusses potential novel metrics that can be used to evaluate metagenome hybrid assembly. It also uses the metrics to evaluate polishing strategies of hybrid assembly. Overall, the manuscript is well prepared, the idea is reasonable and sound, and the technical details are solid.

Experimental design

The experimental design is valid and rigorous. The table and figure quality is excellent.

Validity of the findings

The hypothesis and findings are well reasoned and logical.

Additional comments

I have two major suggestions.

(1) The authors showed that the reference-independent gene count is positively associated with the reference-dependent gene counts. I would like to see the same nice correlation between the reference-independent binning/beta diversity vs reference-dependent diversity (or ground truth). The result can be generated with simulated data or publicly available mock community data.

(2) I would suggest the author conduct the same experiment on low-, medium-, and high-complexity metagenomic data sets. It is well-known that assembly quality may vary a lot depending on community complexity. So, to make the conclusion more convincing and applicable, testing on data with different community complexity levels is highly recommended.

Reviewer 2 ·

Basic reporting

no comment

Experimental design

Typically the benchmarking papers tend to use the datasets with ground truth (e.g. set of reference genomes) known. This way one could evaluate the results precisely without possible issues introduced by imprecise tools like CheckM.

However, this approach for some unknown reason was not used by the authors of the paper and no justification / explanation was done.

There are different approaches that might be used here:
- Simulate the read datasets
- Use well-known mock metagenomes

There are already great datasets available for the second case, e.g. Zymo or Bmock12. This way, the authors could evaluate the results at much better level of accuracy. E.g. CheckM ought to be replaced by Amber, there will be no need to use Prodigal for gene prediction, one could simply use QUAST with gene annotations to estimate the # of complete / fragmented / missed genes.

Also, the authors seem to use only ONT data. It would be great if PacBio data would be analyzed as well and some comparison of the protocol outcomes would be made. I believe both Zymo and BMock12 have both PacBio and ONT data.

Validity of the findings

The authors for some unknown reason only limited themselves to Racon polisher. However, this is not the only polisher for ONT data available. Even more, as some studies show, other tools might perform better, therefore the protocol's results / suggestions definitely might be different.

For example, the authors could refer to [1], evaluate polishing tools mentioned there & compare the conclusions.

1. Lee, J.Y., Kong, M., Oh, J. et al. Comparative evaluation of Nanopore polishing tools for microbial genome assembly and polishing strategies for downstream analysis. Sci Rep 11, 20740 (2021). https://doi.org/10.1038/s41598-021-00178-w

·

Basic reporting

English was quite clear, although I'm not sure about distinguishing correction and polishing - I used to call both long and short read polishing, and authors uses polishing for short and "correction" for long reads

Field background is not sufficient and some really important papers in the field (i.e. https://www.nature.com/articles/s41598-021-00178-w about comparison of different long read polishers or ANY other tools for polishing except racon and pilon) were not cited. And that mention comparison showed that tool used in this study(racon) is not a best option for long read polishing, although tools were compared not on the metagenomes.

It's not clear what is a hypothesis here; speaking on metrics although suggested ones are relevant and potentially useful. Since authors did not tried any other long/short read polishing tool, they should either specify that their "practical recommendations for users" relates only to pilon & racon users, or show that other polishing tools shows the same properties as racon and pilon

Experimental design

The same problem as above can be described as the experimental question definition - do you pretend to show some general properties of long and short read polishing or want to explore the behavior of only two specified tools? Different assembly methods were explored, but polishing (and also basecalling, which also should be important) tools were fixed without any explanation.

Validity of the findings

no comment

Additional comments

also noticed some minor problems: metaSPAdes is refered as a link to webpage but has its own publication; phrase "up to 10 iterations, the greatest number identified among literature and
160 published automated tools" refer to the unicycler paper that likely wasn't meant;

also in other place Unicycler is mentioned as long-read only which is not the true

133 Three common SPAdes-dependent programs were used to generate SR-only, LR-
134 only, and/or SR-first hybrid, i.e., short read assembly followed by connecting contigs using
135 long read alignments, assemblies: metaSPAdes (version 3.15.4)14 known for high-quality
136 metagenomic assemblies, Unicycler (version 0.4.9b)15 with utilities for optimizing SPAdes
137 to enable recovery of circular and high-quality single genomes, and OPERA-MS (version
138 0.9.0)3 capable of automated refinement of high-quality individual genomes within a multi-
139 species metagenome (hybrid only)

---

## Round 0.2 · Minor Revisions

Dear Dr. Smith and colleagues:

Thanks for revising your manuscript. The reviewers are not fully satisfied with your revision. There are issues that still need to be entertained. Please address these ASAP so we may keep progressing with your work.

Please note that reviewer 1 has included a marked-up version of your manuscript.

I understand your arguments (particularly RE benchmarking) and would just ask that you continue to argue the reviewer(s) if you’d rather do that than address changes. Ultimately, I would like to see this criticism alleviated a bit more and have better agreement across the reviewers and yourself.

Good luck with your revision, and please remember to be detailed and in “point-counterpoint” fashion with your next rebuttal.

Best,

-joe

·

Basic reporting

This manuscript discusses potential novel metrics that can be used to evaluate metagenome hybrid assembly. It also uses the metrics to evaluate the polishing strategies of hybrid assembly. Overall, the manuscript is well prepared, the idea is reasonable and sound, and the technical details are solid.

Experimental design

The experimental design is valid and rigorous. The table and figure quality is excellent.

Validity of the findings

The hypothesis and findings are well-reasoned and logical.

Additional comments

I think there could be a misunderstanding of my second comment. I was not asking the authors to benchmark the assemblers under different community complexity. Since the authors proposed several metrics to evaluate assembly quality, the correlation between the proposed metrics and the assembly quality is important. Therefore, it is necessary to evaluate the correlation under different assembly quality levels. For example, the correlation could be strong when the assembly quality is high, and could be weak when the assembly quality is low. Quantifying the effect would help make the contribution of this work clearer.

·

Basic reporting

New title looks quite ambiguous for me.
Although I appreciate change that shows focus on reference-based analysis, optimizing the assembly (at least for me) means optimization of the assembly _process_, that means genome assemblers algorithms and parameter selection. I see that authors means optimization of the final assembly _result_(including polishing and correction) but it is definitely not clear from the title.

I still have problems with the general message of this paper.
Actually I can see three different messages - one how should one assemble, correct & polish hybrid metagenomic dataset(but then extensive benchmarking is definitely required), the second one is how these polished assemblies should be _evaluated_ (to select the best strategy for polishing/correction) , and the third - just a kind of case study description.

From the response to the reviewers and reworded introduction it seems that for the authors the main message is the second one.

So from this point of view, main hypothesis of the paper is formulated at the end of introduction, about the simple reference-free metrics that can be used - present on lines 127-130.

But for a paper about how hybrid genome assembly(+ correction & polishing) should be evaluated, there are so many things missing.

At first, it is completely not clear why when speaking on the evaluation one should limit himself with hybrid assemblies only. "Right" metrics for evaluation should be independent from assembly/polishing/correctness algorithms, and although it keeps away the message of the optimization, I'm pretty sure that described ones can be used for short or long reads only-based workflows.

I was not able to find the complete list of the suggested metrics. See only coding genes count and length, and SR recruitment. Both of theses metrics were suggested before, and speaking on read recruitment counts I found no quantitative justification like correlation with other metrics (although it is evident that more is better)

Tables with all the metrics used for evaluation with the results of the evaluation on the assemblies used could help, but there are just no evaluation tables at all.

Authors shows correlation between a "simple" reference-agnostic metric (total number of sequence in predicted genes) with "complex" one (similarity to the genes in IDEEL DB, fragmented marker genes).
The question is why they are better.
The only motivation I found is on lines 464-465, about computational resources. But in my experience, computational resources required for BUSCO run are incomparably inferior to the resources required to the genome assembly, so this difference can be neglected. If authors feels that computational resources are the main motivation, then a table with CPU hours required for all the steps including assembly, correction, polishing & different methods of evaluation is absolutely required.

Also, for a reference-independent analysis paper a comparison of how this analysis matches the ground truth, reference-based analysis, would be beneficial.

Experimental design

I left the majority of comments on the questions about the experimental design in the text above.
Although I see a clearly formulated research question (in the end of introduction) now, it does not look relevant to some other parts of the paper. It is not clear how this research fills an identified knowledge gap. Authors focused on of evaluation of different stages of iterative polishing, but it is not clear why it should be distinguished with the general reference-free metagenomic assembly evaluation problem.

Validity of the findings

no comment

---

## Round 0.3 · accepted · Accept

Dear Dr. Smith and colleagues:

Thanks for revising your manuscript based on the concerns raised by the reviewers. I now believe that your manuscript is suitable for publication. Congratulations! I look forward to seeing this work in print, and I anticipate it being an important resource for groups studying metagenome assembly. Thanks again for choosing PeerJ to publish such important work.

Best,

-joe